# Mechanistic Insights on Hyperthermic Intraperitoneal Chemotherapy in Ovarian Cancer

**DOI:** 10.3390/cancers15051402

**Published:** 2023-02-22

**Authors:** Olivia G. Huffman, Danielle B. Chau, Andreea I. Dinicu, Robert DeBernardo, Ofer Reizes

**Affiliations:** 1Department of Cardiovascular and Metabolic Sciences, Lerner Research Institute, Cleveland Clinic, Cleveland, OH 44106, USA; 2Division of Gynecologic Oncology, Obstetrics, Gynecology and Women’s Health Institute, Cleveland Clinic, Cleveland, OH 44124, USA; 3Case Comprehensive Cancer Center, Cleveland, OH 44106, USA

**Keywords:** hyperthermia, ovarian cancer, immunity, chemotherapy, HIPEC

## Abstract

**Simple Summary:**

Advanced ovarian cancer is the leading cause of gynecological death with a high rate of reoccurrence indicating the critical need for improved therapeutics. Hyperthermic intraperitoneal chemotherapy (HIPEC) treatment for ovarian cancer has shown efficacy in extending patient overall survival yet the mechanism of benefit remains unknown. The aim of this review is to address the impact of hyperthermia, providing insights into HIPEC efficacy. Here we review reports of HIPEC treatment in ovarian and peritoneal cancers as well as discussion of animal models used for emulating clinical HIPEC.

**Abstract:**

Epithelial ovarian cancer is an aggressive disease of the female reproductive system and a leading cause of cancer death in women. Standard of care includes surgery and platinum-based chemotherapy, yet patients continue to experience a high rate of recurrence and metastasis. Hyperthermic intraperitoneal chemotherapy (HIPEC) treatment in highly selective patients extends overall survival by nearly 12 months. The clinical studies are highly supportive of the use of HIPEC in the treatment of ovarian cancer, though the therapeutic approach is limited to academic medical centers. The mechanism underlying HIPEC benefit remains unknown. The efficacy of HIPEC therapy is impacted by several procedural and patient/tumor factors including the timing of surgery, platinum sensitivity, and molecular profiling such as homologous recombination deficiency. The present review aims to provide insight into the mechanistic benefit of HIPEC treatment with a focus on how hyperthermia activates the immune response, induces DNA damage, impairs DNA damage repair pathways, and has a synergistic effect with chemotherapy, with the ultimate outcome of increasing chemosensitivity. Identifying the points of fragility unmasked by HIPEC may provide the key pathways that could be the basis of new therapeutic strategies for ovarian cancer patients.

## 1. Introduction

Epithelial ovarian, fallopian tube, and primary peritoneal cancers (EOC) are a leading cause of cancer death in women, highlighting the critical clinical need for therapeutic development [1]. The majority (80%) of EOC patients present with advanced stage (III–IV) disease and have a poor prognosis (5-year cancer stage-specific survival 42% and 26%, respectively). Standard of care treatment for advanced EOC involves a combination of debulking surgery and chemotherapy. Hyperthermia has been utilized in cancer treatment for centuries and continues in modern medicine [2]. The therapeutic strategy known as hyperthermic intraperitoneal chemotherapy (HIPEC) in EOC patients at the time of interval debulking surgery (IDS) shows promise as patients experience an extension in overall survival (OS) of nearly 12 months compared to patients undergoing interval debulking surgery (IDS) alone [3]. While this represents the most significant extension of overall survival in EOC patients in over a decade [3], HIPEC mechanisms of action have yet to be understood, thereby limiting further optimization of HIPEC for patients with advanced EOC. Mishra et al. reviewed the history of HIPEC including its adoption in ovarian cancer treatment [2]. Our review focuses on the clinical evidence in support of HIPEC’s benefit in ovarian cancer followed by an analysis of the mechanisms underlying the benefit of hyperthermia in combination with chemotherapy in cancer.

## 2. Hyperthermia in Cancer Therapy—The Clinical Picture

### 2.1. Ovarian Cancer

Epithelial ovarian cancer (EOC) is an aggressive disease of the female reproductive system, often arising from the fallopian tubes, involving the surface lining (epithelial tissue) of the ovaries. A total of 1 in 78 women will experience ovarian cancer in their lifetime [4]. It is expected that more than 22,000 new cases will be reported annually, of which 14,000 will succumb to the disease [5]. EOC has the highest mortality rate of any gynecological cancer with a case-to-death ratio equivalent to lung cancer [6]. Nearly 80% of patients present in late stage (III–IV) thus resulting in poor prognosis [5]. A combination of cytotoxic platinum-paclitaxel-based chemotherapy and debulking surgery remains the standard of care for advanced EOC. While standard treatments have shown initial beneficial outcomes, 70% of patients with advanced disease will experience recurrence within five years, ultimately ending in mortality [7]. The goal of surgery for these patients is to achieve complete macroscopic cytoreduction, as this optimizes overall survival benefit for this intervention [8,9,10]. In patients for whom upfront or primary debulking surgery (PDS) is not safe or complete macroscopic resection is not feasible, neoadjuvant chemotherapy (NACT) followed by interval debulking surgery (IDS) and postoperative chemotherapy allows for initial reduction of disease burden to optimize patients for surgical resection. Randomized clinical trials report no significant difference in progression-free survival (PFS) and overall survival (OS) with this approach compared to primary debulking surgery followed by adjuvant chemotherapy [11]. Despite several new chemotherapy agents demonstrating efficacy against EOC, minimal strides have been made to improve patient OS [11]. The need for new clinical therapeutic strategies is crucial in fighting this disease.

Hyperthermic intraperitoneal chemotherapy (HIPEC) is a promising approach to treating advanced EOC, prolonging the overall survival of patients. HIPEC treatment involves abdominal perfusion of heated chemotherapy via catheter insertion at the time of cytoreductive surgery (CRS) (Figure 1). Perfusion machines maintain a constant infusion temperature through the abdominal cavity. Van Driel and colleagues performed a phase 3 randomized controlled trial (OVHIPEC-1) to test the benefits of HIPEC on newly diagnosed EOC patients, comparing results to treatment without HIPEC [3]. Patients with extensive disease who were not ideal candidates for primary debulking surgery (PDS) or patients with residual tumor after PDS were referred for NACT with or without HIPEC as study participants. Three cycles of NACT were completed prior to entry into the trial. Cytoreductive surgery was completed with or without intraoperative administration of HIPEC using perfusion of cisplatin heated to 40 °C for 90 min via an open abdomen technique. Following surgery, patients in both groups received an additional three cycles of chemotherapy. Results revealed patients receiving HIPEC had an extended OS by nearly 12 months, with no increased rate of adverse effects [3].

To answer the question of whether HIPEC extends patient survival regardless of the timing of cytoreductive surgery, a single-blinded randomized study was performed including patients with stage III or IV ovarian cancer planned for either PDS or IDS [12]. Patients randomized to the HIPEC arm received cisplatin heated to 41.5 °C for 90 min using the closed perfusion Belmont Hyperthermia Pump System. The results reveal an extended PFS and OS in the HIPEC cohort, with an OS increase in 8.2 months in HIPEC patients. Further exploration into any differences between HIPEC at the time of PDS or IDS revealed an increase in PFS and OS in the patients receiving HIPEC after IDS, by 2 and 13 months respectively. Notably, HIPEC at the time of PDS did not extend patient OS and PFS (Table 1). Consistent with Van Driel, these results indicate that HIPEC at the time of IDS prolonged patient survival and improved time to recurrence, providing further evidence of the benefit of HIPEC on extending patient survival against EOC [12].

The standard of care for advanced EOC includes cytotoxic platinum- and paclitaxel-based chemotherapy. In cases of HIPEC, however, single-agent platinum-based chemotherapies, particularly cisplatin or carboplatin, can be used [13]. Several studies have outlined variations in the efficacy of treatment based on the type of chemotherapy utilized in HIPEC. A recent prospective analysis found that PFS was significantly increased with paclitaxel/cisplatin-based HIPEC compared to single-agent cisplatin-based HIPEC [13]. These preliminary findings suggest that the combination of both chemotherapies may be superior to cisplatin alone. Overall survival data is not yet mature. Along the same line, though carboplatin and cisplatin have similar mechanisms of action [13], they can result in different patient outcomes. Zivanovic et al. demonstrated that carboplatin and cisplatin had similar safety profiles in the use of HIPEC for the treatment of recurrent ovarian cancer during secondary cytoreductive surgery [14]. Nevertheless, HIPEC with carboplatin at the time of IDS was not superior to IDS alone in terms of clinical outcomes in this study. These results illustrate that platinum-based HIPEC chemotherapy regimens have varying efficacies, particularly when used alone and when used with additional chemotherapeutic agents.

While the majority of EOC patients initially respond to platinum-based therapy, they often become platinum-resistant (PR) over time, defined as experiencing a disease recurrence within six months of platinum-based therapy [15]. The determination of platinum resistance confers poor prognosis for patients as remaining therapeutic options have limited efficacy. Several studies have suggested that PR patients receiving HIPEC had no alteration in survival rate after HIPEC compared to that of platinum-sensitive (PS) patients [16,17,18]. A randomized study by Spiliotis et al. compared OS in patients undergoing CRS with or without HIPEC for recurrent EOC [16]. Patients who received HIPEC at the time of surgery for recurrence had an OS of 26.7 months compared to 13.4 months for patients who did not receive HIPEC. Furthermore, in the HIPEC group, there was no difference in OS among PS and PR patients (26.8 vs. 26.6 months), while a statistically significant difference in OS was noted between PS and PR patients in the non-HIPEC group (15.2 vs. 10.2 months). This data suggests that HIPEC may overcome the resistance to platinum-based chemotherapy exhibited by the stem cells harbored within recurrent disease [16]. More recently, a retrospective study compared PFS and OS in platinum-sensitive and platinum-resistant EOC patients after cytoreductive surgery (CRS) and HIPEC to determine if CRS with HIPEC in PR patients can overcome PR treatment disadvantages [18]. Patients showed an improved treatment-free interval (TFI) when treated with a combination of HIPEC and secondary CRS, regardless of platinum sensitivity. PS patients had an improved survival to a higher degree than PR patients. Complete tumor resection resulted in significantly increased PFS in PS patients. (Complete cytoreduction was associated with longer survival.) Study limitations included the low number of PR patients and lack of complete resection in nearly half the PR patients. Results suggested that the combination of CRS and HIPEC in PR patients extends the TFI and thus this combination could be a treatment option for patients with PR EOC [18]. Further inquiry is needed to evaluate the role of HIPEC in improving OS for PR patients.

It has been demonstrated that homologous recombination repair (HRR) mutations extend EOC patient PFS and OS [19]. Homologous recombination (HR) is a double-stranded DNA repair mechanism in which damaged chromosomes are repaired and cells are protected from chromosomal aberrations. Disruptions in this pathway result in homologous recombination deficiency (HRD), which impairs a cell’s ability to repair the DNA damaged by chemotherapy [20]. The process of HR includes several mediator genes including BRCA1 and BRCA2; however, these are also among the most mutated HR genes and commonly present in ovarian cancer [21]. Mutations in BRCA1/2 increase the lifetime risk of ovarian cancer development by 40% [22]. Studies show EOC patients with a BRCA mutation have increased chemosensitivity, specifically to platinum-based therapeutics. BRCA mutational status similarly impacts EOC patient response to HIPEC treatment, as hyperthermia impairs the BRCA protein function [23]. An exploratory analysis of the OVHIPEC-1 trial performed by Koole et al. found that patients without BRCA mutations had an increased benefit from HIPEC when compared to those with BRCA mutations [24]. The researchers evaluated tissue samples and tumor DNA from 200 patients with stage III ovarian cancer originally enrolled in the trial and categorized them by BRCA status and HRD status based on copy number variation profile. This study found no significant recurrence-free survival (RFS) benefit or OS benefit to HIPEC among patients with BRCA mutations, HR 1.25 (99%CI 0.48–3.29) and 1.94 (99%CI 0.42–9.16), respectively. Conversely, patients with HRD/BRCA wild-type tumors demonstrated an RFS benefit from HIPEC, HR 0.44 (99%CI 0.21–0.91) without OS benefit 0.55 (99%CI 0.23–1.30). HRD classification may play an increasing role in selecting optimal patients for HIPEC therapy.

The reduction of recurrence seen from HIPEC treatment is promising as the majority of patients with advanced disease experience recurrence within five years [25]. Patients with recurrent disease report a significant impact on their overall quality of life compared to that of women without recurrence, including daily pain, increased emotional burden, activity limitations, and issues concentrating [26]. A single institution cohort study of advanced or recurrent EOC patients receiving CRS and HIPEC was analyzed to identify patterns of recurrence (pelvic, upper abdominal, or extraperitoneal) and whether there exists an association between location of recurrence and patient survival [27]. Results revealed half of the patients analyzed had recurrence outside the peritoneal cavity after HIPEC following CRS. Recurrence location did not impact PFS or OS in HIPEC patients. As HIPEC in ovarian cancer therapy specifically targets the peritoneal cavity, this pattern of spread suggests that HIPEC maintains local control of EOC and may reduce recurrence within the peritoneal cavity [27].

Skepticism surrounds HIPEC as it is perceived to be highly toxic, causing complications [28]. Current HIPEC trials have not reported any adverse effects yet further analysis into patient quality of life post-HIPEC is necessary for the continuation of HIPEC as a safe therapeutic. In a phase-III randomized trial, patients diagnosed with advanced-stage EOC were assessed for any alterations in their health-related quality of life after CRS with and without HIPEC [29]. The study followed patients from before randomization into the trial through 12 months post-treatment including analysis after several rounds of adjuvant chemotherapy. Patient health-related quality of life was assessed via questionnaires at various time points. In patients receiving HIPEC during CRS, no impairment in health-related quality of life was observed. A secondary analysis of PFS and OS confirmed that HIPEC patients after interval CRS had both an extended PFS and OS, consistent with previous findings [3,12].

In summary, an extension in patient survival and reduction in recurrence rate is evident, yet the mechanistic benefit of HIPEC in advanced EOC remains unknown. Studies are highly supportive of the use of HIPEC in the treatment of advanced EOC and indicate the extension of patient survival (Table 1). Based on existing data, the efficacy of HIPEC can be impacted by procedural factors, such as the timing of surgery in the patient’s treatment course and the type of chemotherapy utilized. As previously outlined, different chemotherapy regimens may have altered efficacy when used alone vs in combination with other agents. Similarly, platinum sensitivity is a patient-related factor that affects the utility of HIPEC therapy. Molecular tumor-related factors, including deficiencies in homologous recombination and BRCA status, further influence how patients respond to HIPEC therapy. Additional research evaluating the mechanistic benefits of HIPEC is warranted.

**Table 1 cancers-15-01402-t001:** Summary of clinical findings indicating HIPEC survival benefit.

Author	Year	Study Type	N	Study Details	OS Benefit	PFS Benefit	RFS Benefit
Lim et al. [12]	2022	Single-Blind Randomized	184	HIPEC + interval CRS after NACT in ovarian cancer	13.6 months	2 months	N/A
Ghirardi et al. [23]	2022	Retrospective	70	HIPEC + BRCA mutational status in EOC	No difference between BRCA status	No difference between BRCA status	N/A
Costales et al. [18]	2021	Retrospective	48	PS vs. PR EOC patients given HIPEC after CRS	median 26.9 months in PR patients	N/A	11.2 months in PS patients
Van Driel et al. [3]	2018	Open-Label Randomized	245	Interval CRS ± HIPEC for EOC	11.8 months	N/A	3.5 months
Spiliotis et al. [16]	2015	Open-Label Randomized	120	CRS ± HIPEC for recurrent EOC	13.3 months	N/A	N/A
Safra et al. [25]	2014	Case-Control Study	27	CRS ± HIPEC ± BRCA mutation in EOC	Not reached at time of analysis (70% patients alive)	9 months, no difference in BRCA status	N/A

### 2.2. Additional Applications/Future Directions of HIPEC Therapy

A critical factor in deciding patient eligibility for HIPEC treatment is the presence of peritoneal metastases (PM), which is common among ovarian cancer patients. Pressurized Intraperitoneal Aerosol Chemotherapy (PIPAC) is considered a safe localized treatment for PM. PIPAC is an alternative method of intraperitoneal drug delivery via aerosolized drugs. A prospective PIPAC study enrolled 110 PM patients, 14 of which had a primary ovarian diagnosis, and administered several rounds of PIPAC with or without palliative chemotherapy and bidirectional treatment [30]. The Peritoneal Regression Grading score (PRGS) was utilized to investigate histological treatment response to PIPAC, with a primary outcome of complete or major histological response from three treatments. PIPAC with oxaliplatin or cisplatin and doxorubicin confirmed the primary outcome, PIPAC induced a major or complete histological response, a result independent of patient survival. Quality of life declined post-PIPAC with significantly worsened global health scores despite improvement in fatigue, nausea, constipation, and appetite. PIPAC is known to enhance postoperative pain, yet it cannot be concluded that exacerbated pain is the source of the decline in global health scores [30,31]. PIPAC efficacy warrants additional evaluation for use in primary ovarian cancer patients.

Malignancy is highly reported in primary ovarian cancer patients with a common complication of ascites. Continuous hyperthermic intraperitoneal perfusion chemotherapy (CHIPC) is thought to be advantageous over HIPEC due to the combination of hyperthermia treatment with local chemotherapy via laparoscopic administration [32]. To evaluate CHIPC efficacy in presence of malignant ascites, a 36-patient study was performed, of which 12 patients had primary ovarian cancer [32]. Results reveal successful CHIPC with completely resolved ascites in a majority of patients. No significant adverse effects were reported, and improvement in quality of life was associated with the control of ascites. CHIPC involves the administration of significantly lower doses of chemotherapy compared to systemic treatment, hence the reports of CHIPC being advantageous over HIPEC with respect to the treatment of PM [32].

PIPAC and CHIPC are used as a palliative treatment modality specifically for cancers involving peritoneal metastases. Reports of these therapies being advantageous over HIPEC in cases of primary ovarian cancer with respect to overall survival have yet to be reported.

## 3. Animal Models of HIPEC

An important aspect in elucidating the mechanistic benefit of HIPEC is the development of an animal model to effectively recapitulate clinical HIPEC. Helderman et al. reviewed the current in vivo HIPEC models including the challenges and clinical relevance of each experimental design [33]. Current study designs involve invasive murine models emulating the human surgical technique. Murine models involve either an open or closed perfusion pump system (Figure 2). The open (coliseum) perfusion system involves exposure of the abdominal cavity via a vertical midline laparotomy, securing skin to a ring stand while maintaining sterility. The closed perfusion system involves the introduction of double inflow and outflow catheters through the upper and lower quadrants of the abdomen. Constant temperature is ideally maintained throughout the study duration. Coliseum and closed perfusion systems have shown success in mimicking clinical HIPEC, although neither method of perfusion is without complication. Coliseum perfusion is beneficial as intraoperative organ manipulation is feasible and several studies report total animal survival using the coliseum system; however, reported heat loss limits total clinical recapitulation [34,35]. Simultaneous studies utilizing the closed perfusion system reported a multitude of complications including organ suction into outflow catheters, perfusate leakage, and blood loss at catheter insertion sites [34,35]. Closed and open perfusion systems both permit only one animal treatment at a time, limiting study cohorts to very few animals. Studies report no animal deaths prior to the study endpoint, though most studies follow animals for only days post-HIPEC [36]. Miailhe et al. sought to develop a less-invasive ovarian cancer HIPEC mouse model while limiting complications observed in previous reports [37]. Ten tumor-bearing mice were utilized in a closed perfusion system, in which inflow and outflow catheters were placed at specific locations. A single inflow catheter into the left hypochondria and a single outflow catheter into the left iliac fossa were introduced. Twelve minutes of 43 °C oxaliplatin was infused while mice were kept under constant general anesthesia. All animals survived the duration of treatment with no reported complications. Study limitations include one mouse treatment at a time and the inability to manually stir the perfusate in the abdomen as is possible in the coliseum system. A key component in clinical HIPEC is the combination of CRS prior to HIPEC treatment. The lack of debulking primary tumor in the animals prior to heat is a major study limitation. This improved model of HIPEC showed limited morbidity as only one mouse died prior to the study endpoint [37]. The need for a functional non-invasive animal model for total recapitulation of clinical HIPEC remains, though success in current modalities has reported HIPEC benefit in murine models.

Studies using primarily rats and mice have reported that the HIPEC procedure is possible in animal models, though limited data exist on the mechanistic benefits that HIPEC provides. HIPEC perfusion in colorectal tumor-bearing rats resulted in significantly reduced tumor load in the HIPEC group compared to that of the control and chemotherapeutic-only groups [38]. HIPEC targeting ovarian cancer stem-like cells (CSCs) showed a significant therapeutic effect in immunocompetent mice compared to that of immunodeficient mice [39]. CSCs are a subpopulation of cancer cells exhibiting chemoresistance, thus CSCs may be enriched by chemotherapy [40]. Using the coliseum perfusion system, IP hyperthermia (heated PBS) was infused into the peritoneal cavity for 20 min, maintaining a constant temperature. IP injection of chemotherapeutics was administered immediately after hyperthermia treatment in the treatment group. Mice were then kept under a heat lamp until awake from anesthesia. Results reveal the combination of chemotherapy and IP hyperthermia showed antitumor effects as tumor size was significantly decreased after treatment compared to that of hyperthermia and control groups. Enhancement of antitumor effects was related to the enrichment of chemotherapy by hyperthermia thus reducing the proportion of CSCs in immunocompetent mice. Hyperthermia overcame the chemoresistance, reducing the CSC proportion, in presence of immune system [39].

### Summary of Pre-Clinical Findings and Challenges

Challenges in the study of HIPEC in murine models include the difficulty in recapitulating the clinical HIPEC setting. Clinical HIPEC involves several rounds of neoadjuvant chemotherapy followed by interval debulking surgery and a 90 min heated chemotherapy pumped through the peritoneal cavity. In reported murine HIPEC models, study cohorts are very small due to the nature of the procedure not allowing for multiple animals to be treated simultaneously. Procedure complications have been reported in nearly all cases, including organ suction into outflow catheters, bleeding, and heat loss. Though clinical HIPEC is not completely without complication, heat loss during murine HIPEC poses a major limitation as constant heat is the main premise of HIPEC treatment. The closed perfusion system is a promising model to mimic clinical HIPEC and has successfully shown HIPEC efficacy in reducing murine tumor burden [38,39,41,42]. Studies report the use of heated PBS as IP hyperthermia, though analysis of heated chemotherapeutics would more closely follow human HIPEC. The current unmet need in the current murine models is the low throughput to permit larger cohorts to investigate the impact of the immune system more robustly in HIPEC benefit.

## 4. Mechanisms of Hyperthermia with or without Chemotherapy

### 4.1. Hyperthermic Impact on Chemotherapeutics

Hyperthermic treatment involves the administration of controlled heat above the physiologically normal range, ideally targeting the malignant tissue [43,44]. Studies have demonstrated that hyperthermia enhances the cytotoxic effect of chemotherapeutics when temperatures of 40.5–43.0 °C are applied [43]. The synergistic effect is seen as a linear impact as with increasing temperature, the rate at which cells are killed also increases, notably within platinum-based drugs [43]. Issels et al. reported a comprehensive study of clinical trial results representing standard chemotherapy enhancement by the addition of hyperthermia [45]. The additive effect of hyperthermia on chemotherapy increased median patient survival by over nine years, with a ten-year survival increase of nearly 10% compared to chemotherapy alone [45].

Hyperthermic enhancement of the chemotherapeutic effect may be linked to altered tumor blood flow [44,46]. Blood circulation through the tumor tissue results in enhanced vascular permeability, a physiologically normal pH, and increased oxygenation, thus improving chemotherapy drug distribution throughout the tumor [44]. The molecular mechanism of the synergy between hyperthermia and chemotherapy involves an increase in reactive oxygen species (ROS) with a multitude of downstream effects including an enhancement of drug uptake [47]. An increase in ROS synthesizes DNA damage either directly causing apoptosis or increasing p53 (a gene vital for cell division control and cell death) expression resulting in cell cycle arrest, thus initiating apoptosis [47]. Combined hyperthermia and chemotherapy treatment shows increased apoptotic events via a decrease in heat shock protein (HSP) production, specifically Hsp70 and Hsp90 [47], which are further discussed below in Heat Shock Response.

### 4.2. Heat Shock Response

Stressful conditions including heat shock and tumor presence increase the synthesis of a family of intracellular HSPs [48]. These molecular chaperones are expressed in all cells and are critical for a multitude of functions including protein folding, promotion of immune response, and enhancement of signal pathways essential for cell survival [49,50]. The release of intracellular HSPs in response to heat is dependent on heat shock transcription factor 1 (HSF1), which upon activation by stressors binds to heat shock gene promotors. These extracellular HSPs express pro-immunity function and have been shown to promote antitumor immunity [51]. Extracellular HSPs promote the maturation of dendritic cells (DCs) thus activating the innate immune system [52]. In response to heat shock (42–45 °C), HSPs are released from cells and bind to peptides forming HSP-peptide complexes [53]. The HSP-peptide complexes shuttle antigenic peptides into the major histocompatibility complex (MHC) class I pathway of antigen-presenting cells (APCs) [48] (Figure 3). The MHC-I APC peptide complex binds to the T-cell antigen receptor (TCR) on the surface of T cells, leading to stimulation of the adaptive immune response via activation of CD8+ T cells. CD8+ T cells have shown significant anticancer effects as they produce cytokines targeting tumor tissue.

The picture of the role of HSPs is complex as many cancers exhibit overexpression of Hsp70 and Hsp90, known to be associated with tumor promotion [54]. Due to involvement in multidrug resistance, metastasis, and tumor progression, Hsp90 has been identified as a target for anticancer therapy [54]. Inhibition of Hsp90 stimulates dissociation of HSF1 from Hsp90, activating the heat shock response, with increased expression of heat shock response genes. Simultaneous inhibition of HSF1 is suggested to improve Hsp90 inhibitor anticancer activity due to the HSF1 target genes containing drug resistance and anti-apoptotic properties [54]. Inactivation of Hsp90 increases antitumor immune response thus making Hsp90 inhibitors a promising cancer therapeutic.

Histone deacetylases (HDACs) are enzymes responsible for catalyzing the removal of acetyl functional groups [55]. Deacetylation decreases drug effectiveness, thus HDAC inhibitors are anticancer agents that play a role in the induction of apoptosis and cell cycle arrest [56]. HDAC inhibitors have shown cytotoxic effects on ovarian cancer as HDACs are upregulated after chemotherapy treatment [57]. Sensitizing ovarian cancer cells to Hsp90 inhibitors via histone deacetylase (HDAC) may improve prognosis.

## 5. Hyperthermia Impact on the Immune System

A fever response is a key component to the presence of infection and inflammation and plays a vital role in immune activation, increasing pathogen defense mechanisms [58]. Although HSPs are induced via heat shock, febrile temperatures (38–41 °C) are sufficient to promote HSP production [58]. Clinical results reveal antitumor immunity in the presence of hyperthermia via HSP production and activation of antigen-presenting cells (APCs), resulting in lymphocyte trafficking to the tumor site [59]. Hyperthermia generates the release of HSP-peptide complexes and increases tumor antigens. Febrile temperatures are associated with the activation of circulating neutrophils, which are then recruited to local and distant sites such as tumors, though once temperatures surpass the febrile range neutrophil function will be impaired [58]. The adaptive immune response is heightened during hyperthermia in that NK cells are recruited to the tumor sites under febrile temperatures with enhanced cytotoxicity [58]. The elevated cytotoxicity in NK cells can be linked to increased Hsp70, heat shock protein present in major cellular components, and decreased MHC-I expression by the tumor cells. Tumor cells have upregulated HSP production in response to heat resulting in enhanced antigen-specific cytotoxic T lymphocyte production [58].

The immune system is comprised of two components, innate and adaptive immunity, which work to prevent and limit the invasion of unhealthy cells. The innate immune response is the immediate defense mechanism and provides a general response to foreign substances. The adaptive immune response is a slower, highly specific response that is long-lasting. Immune cells stem from precursor cells found in bone marrow. Myeloid progenitor stem cells are precursors for innate immune cells and include neutrophils, monocytes, DCs, and macrophages. Lymphoid progenitor stem cells are precursors for adaptive immune cells and include B cells, T cells, and natural killer (NK) cells, broadly categorized as lymphocytes. Antigens are foreign substances unrecognizable by the body, thus activating an immune response. Tumors possess a set of specific antigens recognizable by the immune system. APCs at tumor sites uptake the antigens and can create an immune response by activating lymphocytes. Cytotoxic lymphocytes then target tumor cells for destruction. Hyperthermia has the ability to improve this process by the generation of HSPs and activation of APCs, heightening the immune response [59].

The cGAS-STING pathway is an innate immune system component [60] (Figure 4). Hyperthermia has been shown to promote the cGAS-STING pathway in macrophage-like cells [61]. Cyclic GMP-AMP synthase (cGAS) is a protein-coding gene that detects cytosolic DNA and activates the Stimulator of Interferon Genes (STING) pathway with a downstream effect of cytokine activation [62]. DNA is typically localized to the nucleus allowing for control of specialized functions including DNA damage repair and replication [63]. DNA crossing the plasma membrane must translocate across the cytoplasm for nuclear entry through the nuclear envelope [64]. DNA found in the cytoplasm, therefore, is a trigger for immune response activation, as the body recognizes cytosolic DNA as viral entry [65]. Cytosolic DNA is detected resulting in the expression of inflammatory genes, activating defense mechanisms. The cGAS-STING pathway has been discovered to play a vital role in detecting DNA in response to immune defense mechanisms [66]. cGAS interacts with double-stranded DNA (dsDNA) causing DNA ligands to bind with cGAS. Ligand binding induces conformational changes which allow for the catalyzation of ATP and GTP into cyclic GMP-AMP (cGAMP). cGAMP is a second messenger which binds to the surface receptor on the endoplasmic reticulum (ER), activating the Stimulator of Interferon Genes (STING) [67]. STING translocates from the ER to the ER-Golgi intermediate compartments at which TANK binding kinase-1 (TBK1) and interferon regulatory factor 3 (IRF3) are recruited. IRF3 translocates from the Golgi to the nucleus where transcription takes place resulting in the expression of immune-stimulated genes and type 1 interferons. Additionally, STING activates IκB kinase. IκB phosphorylates, mediating the activation of nuclear factor kappa B (NF-κB) activated inflammatory genes including Interleukin 6 (IL-6) and tumor necrosis factor (TNF) [67]. The activation of inflammatory genes elicits an immune response thus hyperthermia is implicated in the promotion of immunity.

## 6. Hyperthermia Impact on Genome Instability

The hallmarks of cancer include genome instability and mutation. Heat causes DNA and protein damage and inhibits cell cycle progression, triggering apoptosis [68]. Hyperthermia induces DNA damage and in combination with chemotherapy has a synergistic effect with chemotherapy increasing sensitivity to chemotherapeutics [69]. Increased chemosensitivity has been attributed to impaired DNA damage repair mechanisms. Chemotherapy alone induces DNA damage, thus in combination with heat, HR is impaired, increasing cancer cell death. To elucidate the effect of hyperthermia on HR, HR-proficient mouse embryonic stem (ES) cells were radiosensitized at normothermic and hyperthermic temperatures and compared to HR-deficient ES cells [70]. Quantification of genes showed that HR-mediated gene targeting had significantly reduced efficiency in ES cells incubated at an elevated temperature. Results suggest hyperthermia inactivates the HR repair mechanism [70] leaving cells reliant on other repair mechanisms such as Poly(ADP-ribose) polymerase-mediated DNA repair.

Poly(ADP-ribose) polymerase (PARP) is a family of proteins involved in DNA repair that when inhibited increases chemotherapy cytotoxicity [71]. PARP enzymes detect single-stranded DNA breaks and bind to the DNA-binding domain. This binding allows the synthesis and transfer of poly(ADP) ribose to acceptor proteins, thus recruiting repair proteins to the site of damaged DNA [58]. Poly(ADP) ribose is involved in the repair of both single-stranded and double-stranded DNA breaks [58]. PARP1 is an enzyme involved in the repair of single-stranded DNA breaks, making PARP1 inhibitors prominent in cancer therapy. BRCA is an HR mediator gene commonly mutated in ovarian cancer. A mutation in BRCA wouldn’t allow for tumor suppression protein release. Hyperthermia on tumor cells resulted in BRCA degradation and HR inhibition [70]. The synergy between hyperthermia and chemotherapy may increase HIPEC benefit for BRCA-positive patients via inhibition of PARP-1-dependent DNA replication [72]. The degradation of BRCA induces increased sensitivity of tumor cells to PARP-1 inhibitors [70]. The studies indicate hyperthermia causes HR-proficient tumors to become sensitive to PARP-1 inhibitors, enhanced by HSP inhibition [70]. The combination of PARP-1 and HSP inhibition with HR inactivation via hyperthermia may be a promising therapeutic in cancer treatment.

## 7. Clinical Correlates on Immune and DNA Repair Activity in HIPEC

Hyperthermia induces an immune response with the downregulation of DNA repair pathways, allowing for tumor suppression. To analyze transcriptomic profile changes induced by HIPEC, pre- and post-HIPEC tumor samples were collected from ovarian cancer patients and compared to normal tissue [73]. HIPEC was given with carboplatin to the four patients included. Samples were analyzed using RNA sequencing. HIPEC induced upregulation of HSPs in tumor tissue with expression changes of Hsp90, Hsp70, Hsp40, and Hsp60 in both normal and tumor tissue. HIPEC with carboplatin induces increased immune-related gene expression in normal tissue with increased protein folding in tumor tissue. Results support the contention that a combination of HIPEC with HSP inhibitors may provide increased therapeutic benefit as some HSPs inhibit protein misfolding thus promoting tumor survival [73].

Tumors of EOC patients receiving HIPEC were collected for whole-transcriptomic analysis to elucidate HIPEC-induced molecular changes [74]. Tumor samples from advanced-stage EOC patients undergoing HIPEC were harvested before and after the procedure. Whole-transcriptomic sequencing, differential gene expression analysis, and gene enrichment analysis were performed. HIPEC induced upregulation of TNFα via the NF-κB pathway [74]. NF-κB is known to be activated through the cGAS-STING pathway and enhanced via hyperthermia [61]. Notably, HIPEC tumors had increased T cell activation as indicated by elevated expression of programmed cell death protein 1 (PD-1), a protein found on the surface of T cells and has a role in immune regulation. PD-1 expression was significantly increased in CD8+ T cells in the post-HIPEC tumor microenvironment. Elevated PD-1 expression post-HIPEC was correlated with improved patient PFS. Post-HIPEC tumors showed an upregulation of immune-related pathways and a downregulation of HR [74]. This analysis references in vitro analyses as preclinical data validation for comparison to human specimens, posing a major limitation of the analyses [74]. Significant elevation in CD8+ T cells, NK cells, and B lymphocyte cells has been observed 30 days after the HIPEC procedure via analysis of peripheral blood in patients with peritoneal metastasis [75]. Results are consistent with other studies showing stimulation of adaptive and innate immune response and inhibition of DNA repair mechanisms via HIPEC.

## 8. Conclusions and Prospects of Future Therapeutic Strategy

Advanced-stage EOC causes more deaths in women than any other gynecological malignancy. Although the standard of care shows initial benefit in treating disease, most patients experience reoccurrence and will ultimately succumb to the disease, indicating a critical need for improved therapy. HIPEC in the treatment of EOC shows significant extension in patient overall survival. Mechanistic insights as to how HIPEC improves patient overall survival provide the opportunity for clinical therapeutic advancement. Hyperthermia induces a multitude of effects making thermotherapy a promising aspect of cancer treatment. Heat activates an immune response, impairs DNA damage repair while inducing DNA damage, and has a synergistic effect with chemotherapy making cancer cells more sensitive to chemotherapeutics. Therapies reliant on DNA damage need also to consider the inhibition of DNA repair mechanisms. In parallel, heat induces the synthesis of HSPs triggering innate and adaptive immunity via the activation of cytotoxic T cells, DCs, and NK cells. Future therapeutic strategies need to include hyperthermic activation of the cGAS-STING pathway, apparently a key component in HIPEC efficacy. Increased cGAS-STING expression would promote increased activation of inflammatory genes leading to increased immune response and targeting the tumor for destruction. Additionally, simultaneous inhibition of Hsp90 and PARP-1 via hyperthermia could sensitize tumors to HR inactivation, impairing tumor cell repair mechanisms. A question for the field is why HIPEC-treated tumors recur. Future studies need to address mechanisms and identify therapeutics to prolong efficacy perhaps by targeting immune surveillance. Animal models have exhibited significant improvement in tumor burden following HIPEC, yet none have reported mechanisms of benefit. HIPEC efficacy is ostensibly reliant on immune system involvement. Studies to elucidate the role of the immune system in HIPEC would provide a starting point for explaining the mechanistic benefit of HIPEC, which could be translated into clinical medicine.

## Figures and Tables

**Figure 1 cancers-15-01402-f001:**
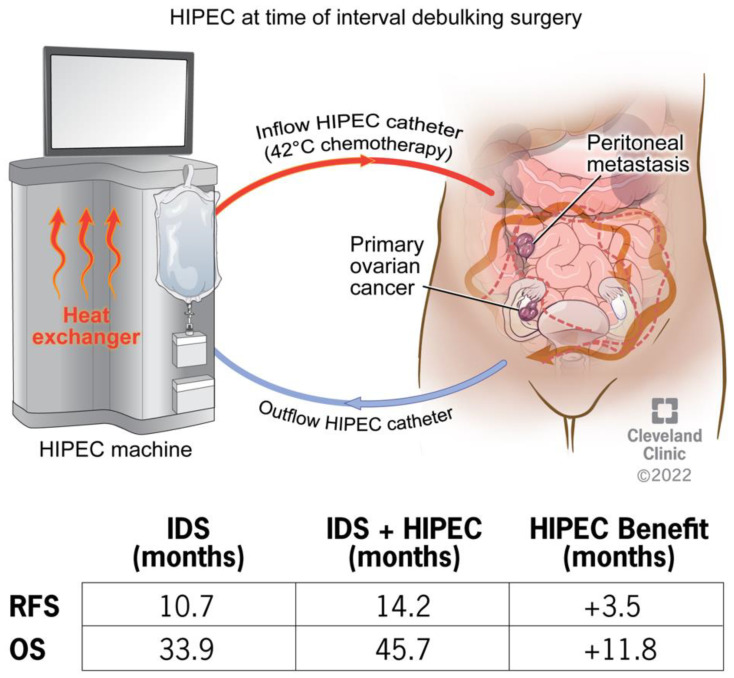
Hyperthermic Intraperitoneal Chemotherapy (HIPEC) treatment in a patient with primary ovarian cancer and peritoneal metastasis. HIPEC has shown an overall survival benefit of nearly 12 months and a regression-free survival benefit of 3.5 months compared to standard of care alone. Reprinted with permission, Cleveland Clinic Foundation ©2022. All Rights Reserved.

**Figure 2 cancers-15-01402-f002:**
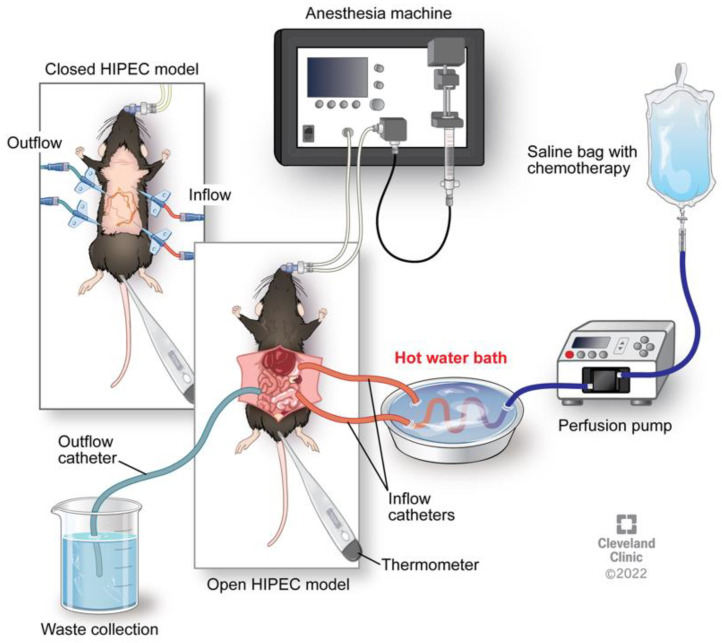
Established murine model of heated chemotherapy treatment. Visualization of the closed system and open (coliseum) system. Reprinted with permission, Cleveland Clinic Foundation ©2022. All Rights Reserved.

**Figure 3 cancers-15-01402-f003:**
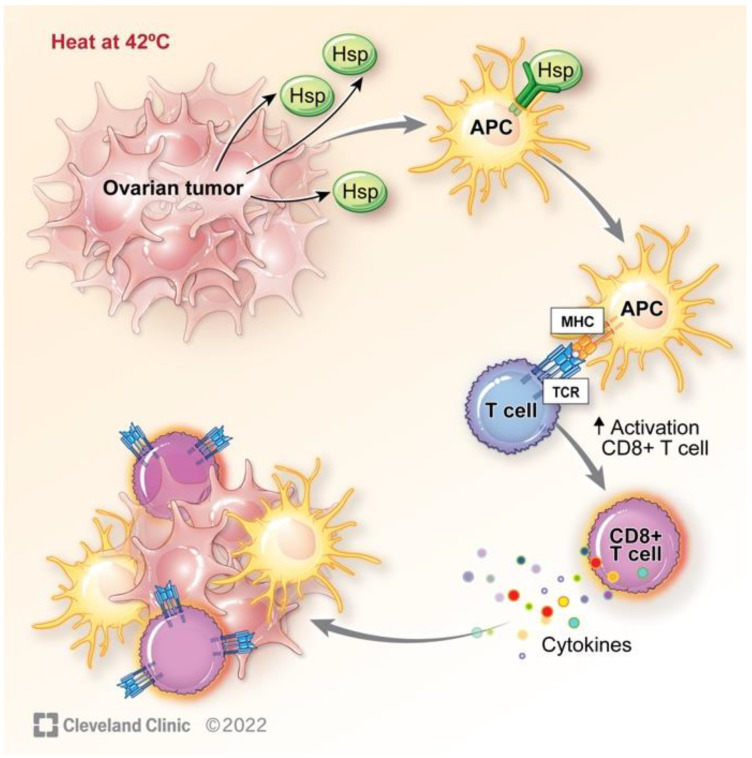
Heat shock protein activation via hyperthermia with downstream cytokine activation, stimulating an immune response. Reprinted with permission, Cleveland Clinic Foundation ©2022. All Rights Reserved.

**Figure 4 cancers-15-01402-f004:**
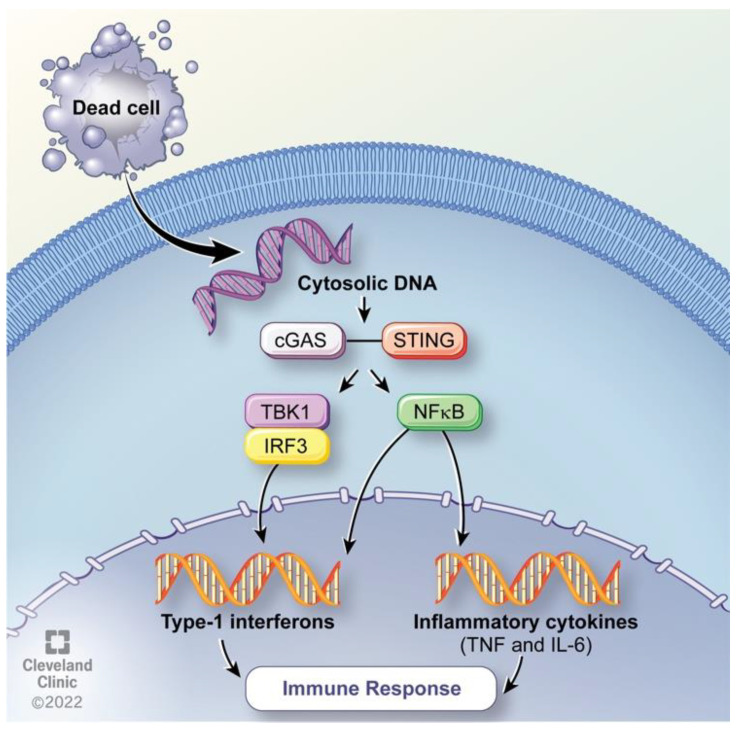
cGAS-STING pathway activation via detection of cytosolic DNA. Reprinted with permission, Cleveland Clinic Foundation ©2022. All Rights Reserved.

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
