# Peer review of "Mechanistic Insights on Hyperthermic Intraperitoneal Chemotherapy in Ovarian Cancer"

_cancers, 2023, doi:10.3390/cancers15051402_

Round 1

Reviewer 1 Report

The authors summarize the role of HIPEC in patients with peritonel metastasized ovarian cancer focussing on the mechanisms of hypertheermia an intraperitoneal chemotherapy. In summary, especially paragrah 2.1 should be revised or deleted and references should be updated as mentioned below.

Comments:

#1) The role of complete macroscopic cytoreduction (CC-0/1) in contrast to debulking surgery should be discussed. It is beyond question that like in most other tumor entities also in patients with pmEOC complete cytoreduction improves survival compared to incomplete cytoreduction/debulking independent of the application of HIPEC. This should be discussed in the revised manuscript and references should be updated regarding this issue.

#2) CRS and HIPEC is standard of care for patients with pseudomyxoma peritonei (PMP) and diffuse malignant peritoneal mesothelioma (DMPM). The HIPEC-treatment of patients with peritoenal metastastasized gastric cancer (pmGC) and pancreatic cancer (pmPC) is under debate. Moreover, there is no consistent data supporting the advantage of CRS/HIPEC in patients with pmGC and especially pmPC (no RCTs, low number of patients). Thus, HIPEC for these tumor entities cannot be considered as a 'standard of care' therapeutic option. These two tumor entities should be removed from paragraph 2.1.

#3) Due to the results of the Fench PRODIGE-7 study the use of HIPEC in patitents with pmCRC is also under debate - especialy regarding the bidirectional Oxaliplatin-based HIPEC regimen. This data should be discussed and references should be updated in the revised manuscript.

#4) HIPEC data for PMP and DMPM should be discussed in paragraph 2.1 and references should be updated (for example PSOGI recommendations).

#5) Table 2 should be limited to the studies regarding pmEOC or removed completely in the revised manuscript.

#6) page 2 line 76: As HIPEC is used for pmEOC patients for more than 30 years the term 'novel' (approach) should be deleted from the revised manuscript.

#7) As the title of the manuscript is 'mechanistic insights' the different HIPEC regimens (used cytostatic agents, time and temperature) and the additive effects of hyperthermia and cytstatic drugs might be discussed in the revised manuscript.

Reviewer 2 Report

Dear Authors,

This article focuses on an important topic related to the clinical implications of HIPEC in ovarian cancer. HIPEC is a method that will be redefined in the therapeutic management of ovarian cancer. As a result, the approach to this topic, although known, continues to present an increased interest that deserves to be analyzed.

Some suggestions could improve the quality of the article:

-    References are in superscript mode.

-   Line 135 to enter the description of the abbreviation for CRS - which is mentioned below in line 171.

-   In line 140, the response to neoadjuvant chemotherapy before CRS is determined by the existence of a subpopulation of malignant stem cells resistant to chemotherapy at the level of the apparently intact peritoneum.

-  To evaluate the role of HIPEC for recurrent platinum-resistant disease with survival benefits.

-     The results of other studies should be discussed, especially the randomized ones, for example, the increase in survival after HIPEC in women with aberrant expression of BRCA1 (OMIM 113705).

-    To specify the situations in which the planned intraperitoneal chemotherapy cannot be completed, such as left colon surgery (low anterior resection).

-   Second-look surgery and HIPEC should be mentioned in managing colorectal peritoneal metastases.

-      Lines 214-220 - what is the role of introducing pancreatic cancer in this section?

-   Line 342 - the initial intervention of sensitization with HDAC - histone deacetylase of ovarian cancer cells to treatment with HSP90i or cisplatin and decreasing the development of resistance to cisplatin may improve the prognosis of ovarian cancer.

-       Approach to peritoneal metastases treated by peritonectomy and HIPEC.

-  In the case of peritoneal metastases, pressurized intraperitoneal aerosolized chemotherapy (PIPAC) can be used as an alternative method for HIPEC.

-       A possible section would be related to metastatic disease and HIPEC.

-      Another topic of discussion would be related to the treatment of malignant ascites by chemotherapy with continuous circulatory hyperthermic intraperitoneal infusion (CHIPC).

-      The effectiveness of HIPEC in the presence of ascites has not been evaluated. For example, ascites do not affect the complete CRS rate and prognosis of patients with massive ascites after HIPEC.

Round 2

Reviewer 2 Report

The objectives proposed by the authors in this study were achieved, and the changes were made according to the recommendations. Overall, this manuscript is well written and documented. Therefore, I recommend for publication of this article.

Kind regards